# Effects of Neuromuscular Training on Physical Performance in Older People: A Systematic Review

**DOI:** 10.3390/life13040869

**Published:** 2023-03-24

**Authors:** Yeny Concha-Cisternas, José Castro-Piñero, Ana María Leiva-Ordóñez, Pablo Valdés-Badilla, Carlos Celis-Morales, Eduardo Guzmán-Muñoz

**Affiliations:** 1Escuela de Kinesiología, Facultad de Salud, Universidad Santo Tomás, Talca 3460000, Chile; 2GALENO Research Group, Department of Physical Education, Faculty of Education Sciences, University of Cádiz, 11519 Puerto Real, Spain; jose.castro@uca.es; 3Instituto de Investigación e Innovación Biomédica de Cádiz (INiBICA), University of Cádiz, 11009 Cádiz, Spain; 4Instituto de Anatomía, Histología y Patología, Facultad de Medicina, Universidad Austral de Chile, Valdivia 5090000, Chile; aleiva@uach.cl; 5Department of Physical Activity Sciences, Faculty of Education Sciences, Universidad Católica del Maule, Talca 3460000, Chile; valdesbadilla@gmail.com; 6Carrera de Entrenador Deportivo, Escuela de Educación, Universidad Viña del Mar, Viña del Mar 2520000, Chile; 7School of Cardiovascular and Metabolic Health, University of Glasgow, Glasgow G12 8TA, UK; carlos.celis@glasgow.ac.uk; 8Human Performance Lab, Education, Physical Activity and Health Research Unit, University Católica del Maule, Talca 3460000, Chile; 9Escuela de Kinesiología, Facultad de Ciencias de la Salud, Universidad Autónoma de Chile, Talca 3460000, Chile; eduardo.guzman@cloud.uautonoma.cl

**Keywords:** postural balance, physical fitness, aged, aging, rehabilitation

## Abstract

This systematic review aimed to assess the available evidence on the effects of neuromuscular training on physical performance in older adults. A literature search was conducted across four databases (Psychology and Behavioral (EBSCO), Scopus, Web of Science and PubMed). The PRISMA guidelines were followed. The PEDro scale and Cochrane risk of bias tool were used to assess the quality of and risk of bias in the studies, respectively. The protocol was registered in PROSPERO (code: CRD42022319239). The outcomes were muscle strength, cardiorespiratory fitness, postural balance and gait speed. From 610 records initially found, 10 were finally included in the systematic review, involving 354 older people with a mean age of 67.3 years. Nine of them reported significant changes in at least one variable related to physical performance in the intervention compared to the control groups. The neuromuscular training caused significant improvements in postural balance, flexibility, cardiorespiratory fitness, strength power of the upper and lower limbs and autonomy. The available evidence indicates that neuromuscular training has a positive effect on some variables of physical performance, especially in postural balance; however, the methodological quality and certainty of the evidence in the available literature are limited. Therefore, a greater number of high-quality studies are required to draw definitive conclusions.

## 1. Introduction

Aging is linked to anatomical and physiological changes capable of inducing a deterioration in functionality and autonomy [1,2]. Among the changes that occur with age, those that affect the musculoskeletal, cardiovascular and somatosensory systems (visual, vestibular and proprioception) are highly prevalent [1]. This causes changes in the ability to maintain posture, gait and the capacity to carry out daily tasks independently [3]. In addition to an increase in muscle weakness, fatigue and functional deterioration, the existing evidence has also suggested that aging is linked to a deterioration of proprioception and the vestibular system, resulting in a poor perception of the position of the body in space, which causes alterations in postural balance, a greater probability of falls and impaired physical performance [4,5].

Physical performance is the capacity to perform daily activities with vigor, without undue fatigue, and with enough energy to take part in leisure activities and handle unexpected emergencies [6,7]. Currently, it is considered a multidimensional concept, operationalized as a set of measurable attributes or components related to health and abilities, including muscular strength, cardiorespiratory fitness, flexibility, postural balance, agility (dynamic postural balance) and gait speed [8]. It has been reported that physical performance is strongly related to health status across the lifespan; however, declines in physical performance can also differ within individuals and therefore relate to healthy aging [8,9].

Various physical exercise programs have been recommended to attenuate the loss of muscle strength and the deterioration of physical performance caused by aging [10,11,12]. For the most part, traditional exercise programs for the elderly are based on interventions with strength training, cardiorespiratory training, combined exercise (cardiorespiratory and strength) and multicomponent programs (cardiorespiratory, strength, flexibility and postural balance) [13,14]. However, the evidence for emerging rehabilitation programs, such as neuromuscular training, is limited, especially in older adults [15,16].

Currently, in comprehensive physical rehabilitation programs, the combination of proprioceptive, strength, and postural balance exercises is known as neuromuscular training [17]. Neuromuscular training is based on the adequate stimulation of peripheral receptors (i.e., mechanoreceptors) in order to improve the integration of muscular responses [18]. According to the literature, neuromuscular training can enhance unconscious motor responses by optimizing afferent signals and the integration of information at the level of the cerebral cortex, favoring dynamic control of the joints [19]. Recently, neuromuscular training programs have been developed in people with knee and hip osteoarthritis (OA) [20,21], in addition to interventions aimed at improving postural balance and the risk of falls in older people. However, the literature is still limited when reporting the benefits that neuromuscular training can generate on physical performance. For this reason, the aim of this systematic review was to analyse scientific articles related to the effects of neuromuscular training on physical performance in older people.

## 2. Materials and Methods

### 2.1. Protocol and Registration

This systematic review was conducted in accordance with the Preferred Reporting Items for Systematic Reviews and Meta-Analyses (PRISMA) guidelines [22]. The protocol was registered and approved by PROSPERO (CRD42022319239).

### 2.2. Eligibility Criteria

For this systematic review, the inclusion criteria were as follows: (i) randomized controlled trials (RCT) or quasi-experimental clinical trials that use neuromuscular training as an intervention in older people of both sexes. 

As the age definitions for “older people” vary around the world, people over 60 years of age were chosen to enable a broad worldwide inclusion. The exclusion criteria were: (i) cross-sectional, retrospective and prospective studies, or whose interventions were not centered on neuromuscular training; (ii) non-original articles (e.g., translations, book reviews, letters to the editor); (iii) duplicate articles; (iv) review articles (e.g., narrative reviews, systematic reviews, meta-analyses); and (v) case studies. Additionally, the studies were incorporated into the systematic review using the PICOs framework (population, intervention, comparator, outcomes and study design).

### 2.3. Information and Database Search Process

The search process was performed between May and July 2022, using four databases: Psychology and Behavioral (EBSCO), Scopus, Web of Science and PubMed. Original articles without language restrictions and published in the last 60 years (January 1950 to the date of extraction April 2022) were included. The search string used was the following: (“neuromuscular training” OR “proprioceptive training program” OR “sensorimotor training program”) AND (“physical performance” OR “physical fitness” OR “functional fitness” OR “functional capacity” OR “functionality” OR “performance” OR “strength” OR “resistance” OR “endurance” OR “cardiorespiratory fitness” OR “balance” OR “stability” OR “agility” OR “postural control” OR “gait” OR “speed” OR “locomotion” OR “SPPB”) AND (“old people” OR “elders” OR “senior” OR “old adult” OR “aged” OR “older people” OR “older adults” OR “geriatric”).

### 2.4. Studies Selection and Data Collection Process

The research was transferred to the EndNote references manager (version X8.2, Clarivate Analytics, Philadelphia, PA, USA). The searching, removing duplicates, screening titles and abstracts, and examining full texts were performed by two independent authors (YCC and EGM). At this point, no discrepancies were discovered. The full text of all the potentially eligible studies was reviewed, and the reasons for excluding those studies that did not meet the selection criteria were reported.

### 2.5. Methodological Quality Assessment and Risk of Bias

The selected studies were evaluated through the PEDro scale. This scale evaluates the methodological quality of the studies, considering 11 points that include the blinding procedure, statistical analysis, information on randomization and the presentation of the results in the evaluated research [23]. Criterion 1 assesses the external validity and is not included in the result. From criteria 2 to 11, the internal validity of the article was evaluated with a standardized scoring system (range from 0 to 10). The study quality was classified as excellent (9–10 points), good (6–8 points), fair (4–5 points) and poor (<4 points) [24]. This process was conducted independently by two authors (YCC and EGM).

Using Review Manager 5.3 software, two independent authors (YCC and EGM) evaluated the risk of bias in the included trials (Cochrane Collaboration, Oxford, UK), and differences concerning the methodological quality were resolved by discussion. The Cochrane criteria were used to assess the risk of bias, and selection bias, performance bias, detection bias, attrition bias, reporting bias and other potential biases were considered. The risk of bias was categorized into three grades: low risk, high risk and some concerns [25]. 

### 2.6. Data Synthesis

The following information was obtained and examined from the selected studies: (i) authors and publication year; (ii) the country where the research was carried out; (iii) the study design; (iv) the sample’s initial health condition; (v) the number of participants in the intervention and control groups, and percentage of women; (vi) the mean age of the sample; (vii) the activities developed in the experimental groups (EG) and control groups (CG), the training volume (total duration, weekly frequency and time per session); (viii) the data collection instruments of physical performance; and (ix) the main outcomes of the studies.

## 3. Results

### 3.1. Studies Selection

In the study identification phase, 610 articles were found. Then, the duplicates were removed (*n* = 493), and the studies were filtered by selecting the title, abstract, and keywords, obtaining 117 references. In the analysis phase, two studies were excluded because the full text was not accessed. Forty-three studies were analysed in the full text, four of which were excluded due to not being neuromuscular training; 13 because the mean age of the participants was less than 60 years; 3 for not having physical performance assessments; 8 for not having a control group and 5 for being case studies or reviews. After this process, 10 studies in total met all the requirements for selection (Figure 1) [26,27,28,29,30,31,32,33,34,35].

The PEDro scale was used to evaluate the 10 selected studies (Table 1). All the studies achieved a score equal to or greater than five points on the scale and were classified as fair: 5/10 [28,29,31,34] and good: 6/10 [26,27,32,33], 7/10 [30] and 8/10 [35]. No studies of excellent methodological quality were found.

### 3.2. Risk of Bias

The details about the risk of bias in the included studies are shown in Figure 2A,B. Five studies presented a high risk of selection bias, as they used single-blinded assessments and did not perform adequate randomization, while five were categorized as “some concerns”.

### 3.3. Studies Characteristics

Table 2 summarizes the variables analyzed in each of the selected studies. Of these, two were performed in South America [28,32], five in Europe [29,30,31,33,34], two in Asia [26,35] and one in Africa [27]. Concerning the design of the studies, eight were randomized controlled trials, and two were quasi-experimental clinical trials.

### 3.4. Sample Characteristics

Seven studies had between 20 and 40 participants [26,27,28,29,30,32,35] and three between 41 and 70 [31,33,34], totaling a sample of 354 older people (EG = 171; CG = 183) with a mean age of 67.3 years (EG = 67.8 years; CG = 66.8 years). On the other hand, two studies reported that the older people included were people with no apparent health problems [29,31]; three reported older people with sedentary lifestyles [28,32,33], three studies reported knee OA [27,30,35] and one included older people with peripheral neuropathy [26]. Only one study involved post-menopausal women [34]. 

### 3.5. Physical Performance Outcomes and Collection Instruments

Muscle strength was assessed by two studies [32,35]. The first assessed upper and lower limb strength through functional tests included in the Senior Fitness Test battery [32], while the second study measured lower limb muscle strength with the climbing-up-and-down-stairs test [35].

Cardiorespiratory fitness was analysed by one of the studies using the six-minute walk test [32]. For its part, the flexibility of the lower and upper limbs was assessed using the Sit and Reach and Back Scratch Tests, respectively [32]. 

Six studies evaluated postural balance through the use of the Biodex Stability System (BBS) [27,30] or force platforms [26,31,33,34] by analyzing the stability indices and variables of the center of pressure (COP), respectively.

Also, functional tests were used to assess postural balance. Some studies used the Timed Up and Go test (TUG), Functional Reach Test (FRT) and One-Legged Stance (OLS) Test [26], while Martínez-Amat et al. (2013) included the Berg balance scale and the Tinetti scale [31]. 

On the other hand, Esposito et al. (2021) incorporated the test as an evaluation Four Square Step Test, a test where the person was required to step sequentially over four rods arranged in a cross on the ground [29]. 

Finally, three studies included the evaluation of gait speed, for which they used a 10 m walk test [32] and a 60 m walk test [30,35].

### 3.6. Interventions and Dosage

All the studies had two groups of analysis, the EG, whose participants took part in the intervention with neuromuscular training [26,27,28,29,30,31,32,33,34,35], and the CG, whose participants developed other activities and habitual activities. 

The neuromuscular training carried out in the studies included proprioceptive, strength and postural balance exercises. Of the 10 studies selected, eight maintained the classic structure of a training session, including a warm-up, development (where neuromuscular activities were performed) and a cool-down; however, not all reported all the activities performed and/or the training volume [27,28,31,32,33,35]. 

Table 3 summarizes the neuromuscular training protocols used by the studies. Five studies included proprioceptive exercises that were progressive in intensity, beginning with simpler activities such as (a) standing upright on a firm surface and then on a soft surface; and (b) a one-legged stance with eyes open and closed on a firm surface and then on a soft surface [26,27,28,30,34]. More complex functional activities were also incorporated, such as: (a) walking on a firm surface and then on a foam surface; (b) squat or postural balance exercise on a wobble board [27,28,34]. Agility was also part of the intervention, including walking activities in different directions [28]. The routines included activities aimed at developing coordination, speed and muscle power, as well as multi-articular exercises for the upper and lower limbs [32,33]. Among the materials used for the execution of the activities were firm mats (yoga mats), soft mats, oscillating tables, air pillows and cones [32], in addition to softballs, tennis balls, rubber bands [27], BOSU balls [31] and a suspension system [35]. 

Regarding the duration of the interventions, three lasted six weeks [27,30,33], four lasted 8 weeks [26,28,32,35], two lasted 12 weeks [29,31] and one lasted 36 weeks [34]. Regarding the training frequency, all the studies reported a weekly distribution of two or three sessions. The minimum duration of a session was 15 min [34], while one study reported a duration of 80 min [26]. 

For the CG, in four studies the participants maintained their activities of daily living [28,29,30,31]. In two studies, the older people participated in physical fitness programs, which focused on exercises and activities to develop muscular endurance and strength, cardiorespiratory fitness, flexibility, agility, and postural balance [27,32]. Other investigations incorporated a routine of physical therapy including thermotherapy, interferential therapy and instructions for exercise at home [35], and the participants in the study by Stolzenberg et al. (2013) were trained on a vibrating platform [34]. Finally, Ahmad et al. (2019) considered education on diabetic foot care as an intervention for the CG [26], while, in the study by Rezaeipour & Apanasenko, the participants had to remain seated for a period of time per week (60 min) [33].

### 3.7. Main Outcomes

The results of this systematic review show that neuromuscular training has beneficial effects on physical performance in older people. Of the 10 articles included in the review, nine of them reported significant changes in favor of the EG in at least one variable related to physical performance [26,27,28,29,30,31,32,33,34] compared to CG.

It was shown that neuromuscular training caused significant changes in postural balance in all the studies that included this variable as an outcome [26,27,29,30,31,32,33,34]. One of the postural balance variables where a beneficial effect was evidenced was the COP, both with eyes open [26] and closed [26,34], which indicates a better postural balance. Specifically, the COP variables that decreased with the neuromuscular training were the COP sway [26], COP velocity [26,33], COP total mean velocity [33] and Romberg speed [31], which showed significant changes in the anteroposterior (AP) [26] and mediolateral (ML) directions [26,33]. The studies showed a decrease in the overall stability index [27,30], ML and AP stability indices [27] after the neuromuscular training. In addition, favorable balance changes were exhibited in the neuromuscular training groups in clinical postural balance tests such as FRT [26], TUG [26,33], OLS [26], Tinneti scale [31], Berg balance scale [29,31] and Four Square Step Test [29]. 

Finally, it was possible to determine the favorable effects of neuromuscular training on flexibility [32], cardiorespiratory fitness [32], strength and power of upper and lower limbs [32] and on functional autonomy [28]. 

## 4. Discussion

The objective of this systematic review was to analyse the scientific evidence regarding the effect of neuromuscular training on physical performance in older people. The main findings show that neuromuscular training has a positive and significant effect on some variables of physical performance, especially in postural balance. After reviewing 610 records, ten studies met the inclusion criteria; however, the quality of the research was only categorized as fair and good, with no excellent-quality studies. In terms of the risk of bias, five studies were categorized as high risk, while the remaining five had some concerns.

Among the main results of this systematic review is that seven of the articles included reported that a neuromuscular training program improves postural balance in older people, which is expressed in improvements in the stability indices, such as the COP variables [26,34], in functional tests, such as the TUG [26,33], OLS [26], FRT [26], and in the score of postural balance scales, such as the Berg scale [29,31] and Tinetti [31]. These results are in line with previous systematic reviews, where it has been reported that older women with osteoporosis who execute a neuromuscular training program improved static and dynamic postural balance, in addition to increasing the speed of motor responses [36]. Therefore, it has been suggested that this type of intervention would help improve the performance of activities of daily living and reduce the frequency of falls in older people [36].

When performing an exhaustive search of the literature, extensive benefits of neuromuscular training are reported, not only in functionality but also in the health and quality of life of people. However, neuromuscular training is still used more frequently in athletes and young people; for example, Vásquez-Orellana et al. (2022) demonstrated that a six-week neuromuscular training program improves postural balance and proprioception in basketball players with functional ankle instability [37]. For their part, Guzmán-Muñoz et al. (2020) reported improvements in static and dynamic postural balance in obese children after only four weeks of neuromuscular training [38]. It has been suggested that neuromuscular training can enhance postural balance by stimulating anticipatory postural adjustments [39]. The repeated exposure to stability challenges on stable and unstable surfaces generates feedforward adjustments that would promote appropriate muscle activation [40]. Additionally, neuromuscular training could stimulate proprioception by inducing peripheral and central neural adaptations [17]. The central nervous system can use ascending information from the tendons (Golgi tendon organs), muscles (muscle spindles) and other mechanoreceptors (located in the ligaments, capsule and skin) to construct precise voluntary movements or to correct disturbances [41]. Consequently, the peripheral adaptations that may have occurred because of neuromuscular training likely resulted from the repetitive stimulation of the articular mechanoreceptors on both stable and unstable surfaces and in the eyes open and eyes closed conditions [38]. Therefore, neuromuscular training is mainly based on providing adequate sensory information to peripheral receptors so that the integration of muscular responses is more efficient, combining proprioceptive, strength and postural balance exercises [17,18]. This adjustment in motor responses would not only allow improvements in postural balance but also in other components of physical performance reported by the authors included in this review. In this way, it is suggested that neuromuscular training be used both in rehabilitation programs and in the prevention of injuries in older people [29]. The adaptations induced by neuromuscular training could be beneficial not only for athletes and young people, which suggests the need to develop studies that include this training methodology in the elderly population, given the anatomical, physiological and functional changes that occur with age.

On the other hand, two of the ten articles selected in this systematic review showed that older people who performed neuromuscular training improved performance in functional tests that assess muscle strength of upper and lower limbs compared to people from the CG [32,42]. In general, it has been established that older people have reduced muscle strength due to the anatomical changes that occur in the musculoskeletal system and the deterioration of sensory discrimination capacity [28]. Based on these antecedents, various authors have postulated that these deficiencies must be addressed specifically, where physical activity programs must not only be oriented towards isolated work on physical qualities but must include mixed activities, necessarily incorporating strength exercises, proprioception and postural balance—activities considered as the basis within a neuromuscular training program [43,44].

Finally, only one of the three studies selected for this review that included gait speed among its functional variables reported positive changes in favor of older people who performed neuromuscular training [30]. The divergence in the results found by the studies could be attributed to the heterogeneity of the tests used to assess walking speed, the differences in the distance traveled (60 m walk test and 10 m walk test), the different methodologies used to execute the test (normal walking speed or fast walking speed) or due to the different functional conditions and/or diseases that the older people in the studies had (sedentary older people and knee OA). Previous research has indicated that the discrepancy between the distance traveled and the pace of walking that is requested of the participants affects the results of the test [45]. Similarly, it remains unclear whether a longer walking distance yields a more accurate determination of gait speed than a shorter distance [45]. Regarding the speed that should be requested to perform the walk test at a normal pace, it seems to be more adequate to estimate the physical function and deterioration due to chronic disease [46]. Mobility is one of the most important aspects of physical function, especially as a prerequisite for performing activities of daily living. Given that walking is a complex motor phenomenon, resulting from the interaction of the nervous, musculoskeletal, vestibular and somatosensory systems, it is important to develop physical activity interventions that incorporate different motor skills with the aim of promoting the autonomy of older people.

### Limitations

The main limitation of our systematic review is the low quality of the evidence found and the high risk of bias in the articles included. It is also worth mentioning the limited number of studies available, in which the participants were mainly women, and the heterogeneity found between the studies in terms of interventions (e.g., participants with different health statuses, duration of the session or sessions per week and the diversity of measurement instruments used). More evidence is needed to determine the real effects of neuromuscular training on the variables that make up physical fitness (cardiorespiratory fitness, muscle strength, flexibility, postural balance and agility) and to confirm the clinical relevance of these training-induced improvements.

In the same way, it is possible to recommend future studies with higher methodological quality, which include, among other aspects, double-blind evaluations and improvements in the randomization process of the sample, with the intention of clearly reporting the methodology used for the assignment of the participants to the different working groups. Finally, a strength of the present review was the strict inclusion/exclusion criteria (e.g., analyzing RCTs and quasi-experimental clinical trials).

## 5. Conclusions

Despite the limitation represented by the methodological quality of the available intervention studies, the evidence collected indicates that neuromuscular training has a positive and significant effect on some variables of physical fitness, especially in postural balance, so this type of training is seen as an effective resource to improve the functionality of the elderly and mitigate the deleterious effects caused by aging.

## Figures and Tables

**Figure 1 life-13-00869-f001:**
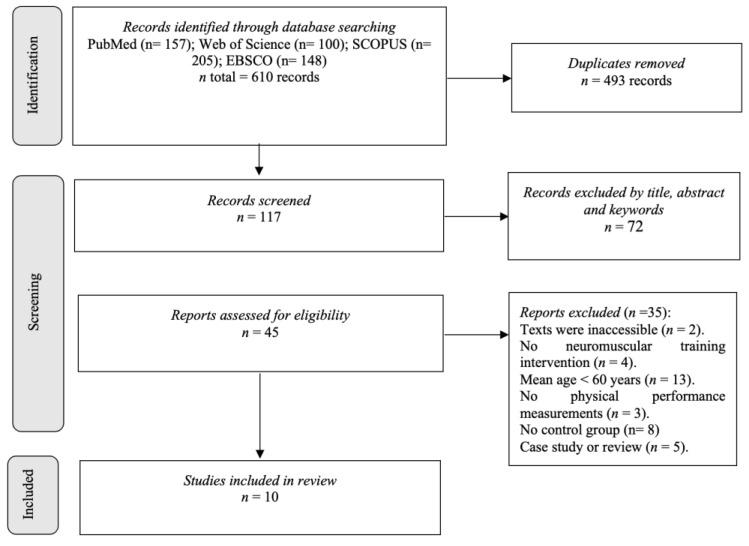
PRISMA flowchart of the search strategy and study selection.

**Figure 2 life-13-00869-f002:**
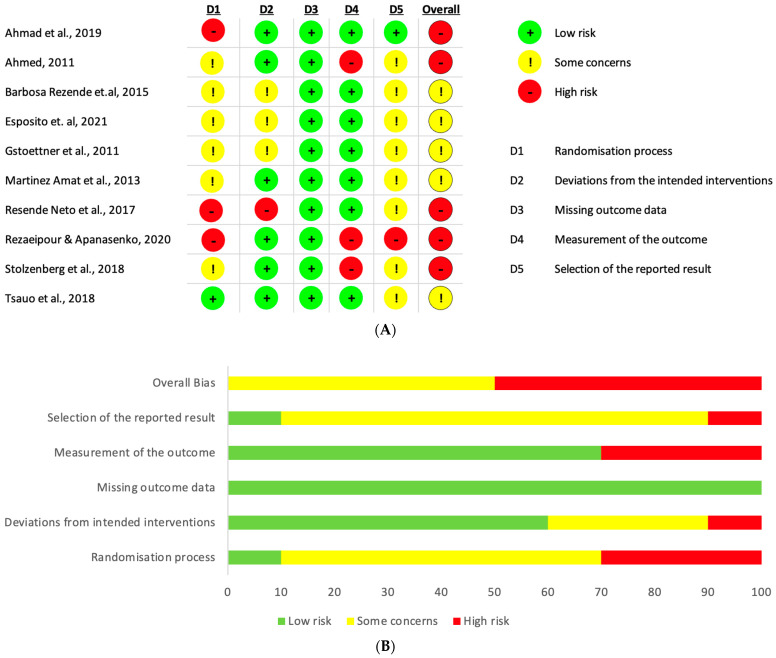
Assessment of risk of bias based on the Cochrane risk of bias tool. (**A**) Risk of bias graph [26,27,28,29,30,31,32,33,34,35]; (**B**) risk of bias summary.

**Table 1 life-13-00869-t001:** Study quality assessment according to PEDro scale.

Study	Ahmad et al., 2019 [26]	Ahmed, 2011 [27]	Barbosa Rezende et al., 2015 [28]	Esposito et al., 2021 [29]	Gstoettner et al., 2011 [30]	Martinez-Amat et al., 2013 [31]	Resende Neto et al., 2017 [32]	Rezaeipour & Apanasenko, 2020 [33]	Stolzenberg et al., 2018 [34]	Tsauo et al., 2008 [35]
1—Inclusion criteria *	Yes	Yes	Yes	Yes	Yes	Yes	Yes	Yes	Yes	Yes
2—Random allocation	Yes	No	Yes	Yes	Yes	No	Yes	Yes	Yes	Yes
3—Concealed allocation	No	No	No	No	Yes	No	No	No	No	No
4—Group similarity at baseline	Yes	Yes	Yes	Yes	Yes	Yes	Yes	Yes	Yes	Yes
5—Blinding of participants	No	No	No	No	No	No	No	No	No	Yes
6—Blinding of therapists	No	Yes	No	No	No	No	No	No	No	Yes
7—Blinding of assessors	No	No	No	No	No	No	No	No	No	No
8—Outcome measures in 85% of sample	Yes	Yes	No	No	Yes	Yes	Yes	Yes	No	Yes
9—Intention-to-treat analysis	Yes	Yes	Yes	Yes	Yes	Yes	Yes	Yes	Yes	Yes
10—Comparison between groups	Yes	Yes	Yes	Yes	Yes	Yes	Yes	Yes	Yes	Yes
11—Measures of central tendency and dispersion	Yes	Yes	Yes	Yes	Yes	Yes	Yes	Yes	Yes	Yes
Total score	6/10	6/10	5/10	5/10	7/10	5/10	6/10	6/10	5/10	8/10

* Criterion not considered for the total score.

**Table 2 life-13-00869-t002:** Studies reporting on the effect of neuromuscular training on physical performance in older people.

Study	Country	Study Design	Sample’s Initial Health	Groups (*n*) and Sample Size Female (%)	Mean Age (Year)	Intervention (s)		Data Collection Instruments of Physical Performance	Main Outcomes
						Experiment Group (EG)	Control Group (CG)		
**Ahmad et al., 2019** **[26]**	India	RCT	Diabetic peripheral neuropathy	21 EG: 12 CG: 9 33.3% Female	66.7 64.7	Neuromuscular training and foot care. 3 × 80 min/week 8 weeks	Diabetes education and foot care.	FRT (cm) TUG (sec) OLS (sec) Balance in Force Platform (COP sway (open eyes) and COP sway (closed eyes)).	EG: ↑ FRT, ↑ OLS (closed eyes), ↓ TUG, ↓ COP sway AP (open eyes), ↓ COP sway AP *closed eyes), ↓ COP sway ML (closed eyes). EG vs. CG: significant differences in favor of EG.
**Ahmed, 2011** **[27]**	Egypt	RCT	Knee OA	40 EG: 20 CG: 20 100% Female	60.0 62.0	Traditional exercise program in addition to neuromuscular training 3 × 30 min/week 6 weeks	Traditional exercise program	Balance in Biodex Stability System (BBS) (overall stability index, AP stability index and ML stability index).	EG: ↓ overall stability index, ↓ ML stability index, ↓ AP stability index. EG vs. CG: significant differences in favor of EG.
**Barbosa Rezende et al., 2015** **[28]**	Brazil	RCT	Sedentary lifestyle	30 EG: 15 CG: 15 100% Female	NRS	Neuromuscular training 3 × 30 min/week 8 weeks	Usual activities	Functional autonomy (10 m walk, getting up from a sitting position, getting up from the prone position, getting up from a chair and moving around the house, GDLAM index).	EG: ↓10 m walk, ↓ getting up from a sitting position, ↓ getting up from the prone position, ↓ getting up from a chair and moving around the house, ↓ GDLAM index) in favor of EG. EG vs. CG: was not reported
**Esposito et al., 2021** **[29]**	Italy	RCT	Apparently healthy	30 EG: 15 CG: 15 60% Female	NRS	Neuromuscular training 2 × 60 min/week 12 weeks	Not reported	Balance: Berg balance scale (score) Four Square Step Test (sec).	EG: ↑ Berg balance scale and ↓ Four Square Step Test. EG vs. CG: significant differences in favor of EG.
**Gstoettner** **et al., 2011** **[30]**	Austria	RCT	Knee OA	38 EG: 18 CG: 20 79% Female	72.8 66.9	Neuromuscular training 3 × 30 min/week 6 weeks	Usual activities	Balance in BBS (overall stability index, AP stability index and ML stability index). Gait speed (60 m walk test).	EG: ↓ overall stability index. EG vs. CG: significant differences in favor of EG in AP stability index
**Martinez-Amat et al., 2013** **[31]**	Spain	Quasi-experimental	Apparently healthy	44 EG: 20 CG: 24 43% Female	79.3 77.0	Neuromuscular training 2 × 50 min/week 12 weeks	Usual activities	Balance in Force Platform (AP displacement and ML displacement (open eyes and closed eyes), COP speed and Romberg quotient (open eyes and closed eyes). Berg balance scale (score) Tinetti scale (score).	EG: ↓ Romberg speed, ↑ Tinetti scale and ↑ balance Berg scale. EG vs. CG: significant differences in favor of EG in ML displacement (open eyes), AP displacement (open eyes), AP displacement (closed eyes), Romberg speed, Tinetti and Berg balance scale.
**Resende Neto** **et al., 2017** **[32]**	Brazil	RCT	Sedentary lifestyle	32 EG: 16 CG: 16 100% Female	64.6 66.6	Neuromuscular training 3 × 60 min/week 8 weeks	Traditional training	Physical performance (sit and reach, back scratch, TUG, sit to stand, elbow flexion, six-minute walk). Dynamic strength (supine, rowing and squatting) Muscle power (supine, rowing and squatting).	EG: ↓ sit and reach, ↓ back scratch, ↓ TUG, ↑ sit to stand, ↑ elbow flexion, ↑six-minute walk. ↑ Dynamic strength (supine, rowing and squatting) ↑ Muscle power (supine, rowing and squatting). EG vs. CG: TUG, sit to stand, elbow flexion and six-minute walk in favor of EG.
**Rezaeipour & Apanasenko, 2020** **[33]**	Ukraine	Quasi-experimental	Sedentary lifestyle	48 EG: 24 CG: 24 100% Female	70.1 69.3	Neuromuscular and proprioceptive training 3 × 60 min/week 6 weeks	Seated Rest	Balance in Force Platform (COP total mean velocity, COP velocity AP and COP velocity ML (open eyes and closed eyes)).	EG: ↓ COP velocity ML (closed eyes), ↓ COP total mean velocity. EG vs. CG: significant differences in favor of EG.
**Stolzenberg et al., 2013** **[34]**	Germany	RCT	Post-menopausal women	68 EG: 31 CG: 26 100% Female	67.3 65.9	Neuromuscular training 2 × 15 min/week 36 weeks	Whole-body vibration	Balance in Force Platform (COP velocity and COP area). Balance on unstable surface (velocity of movement mm/s).	EG: ↓ velocity of movement and ↑ COP velocity semi-tandem (closed eyes). EG vs. CG: No differences.
**Tsauo et al., 2008** **[35]**	Taiwan	RCT	Knee OA	29 EG: 15 CG: 14 100% Female	61.7 60.1	Physical therapy program in addition to neuromuscular training 3 × 30 min/week 8 weeks	Physical therapy program and instructions for exercise at home	Physical performance (60 m walk test, Figure-of-8 walk test, climbing up and down stairs)	EG: no changes significant. EG vs. CG: No differences.

AP: anteroposterior; BBS: Biodex Stability System; CG: control group; cm: centimeters; COP: center of pressure; EG: experimental group; FRT: functional reach test; GDLAM: Latin American Development for the Elderly Group. ML: mediolateral; mm/millimeter/second; NRS: not reported separately; NRCT: non-randomized controlled trial; RCT: randomized controlled trial; OA: osteoarthritis; OLS: One-leg stance; sec: seconds; TUG: Timed Up and Go test; ↑: significant increase; ↓: significant decrease.

**Table 3 life-13-00869-t003:** Neuromuscular training protocol.

	Neuromuscular Training
Ahmad et al., 2019 [26]	Warm-up: used cycle ergometer or treadmill at the intensity of 50% to 60% HRmax (10 min). Developing: wall slides, core exercises, balance exercises on an unstable surface, and gait training (different patterns of walking) (50 to 60 min). Cool-down: included deep breathing, abdominal breathing and mild stretching (10 min).
Ahmed, 2011 [27]	Consisted of three stages: static, dynamic and functional (30 min). First phase (static): Standing upright position on a firm surface, then on a soft surface; Single Leg Stance with closed eyes (first the affected limb, then the non-affected limb) on a firm surface, then on a soft surface, half-step position, one-leg balance. Second phase (dynamic) in addition: Forward-stepping lunge and T-band kick exercises. Third phase (functional) in addition: Walking exercise on a firm surface, then on a foam surface, squatting exercise, balance exercise on a wobble board. Warm-up and cool-down were not reported.
Barbosa Rezende et al., 2015 [28]	Developing: Consisted of eight stages, which included activities such as: walking over 35 cm wide cylindrical blocks, stepping over five 0.75 cm high signal cones, walking in a straight line over 3 m of mats placed end-to-end on the floor, moving forwards and sideways to negotiate obstacles, among others (30 min). Warm-up and cool-down were not reported.
Esposito et al., 2021 [29]	Warm-up: joint mobilization (6 min). Developing: circuit with five exercises: balance on one leg, getting up from a 60 cm high chair, walking on a proprioceptive pad, walking in a straight line of 10 ms and throwing softballs toward a wall. Cool-down: walking, stretching and mobilization (50 to 60 min).
Gstoettner et al., 2011 [30]	Warm-up: walking on heels and toes and brisk walking (5 to 10 min). Developing: Proprioception and balance training were carried out barefoot on different mats and included exercises with eyes open and repeated with eyes closed. The program included four exercises: slide/step forward/backward, step forward/backward, Single Leg Stance and squats. (30 min). Cool-down was not reported.
Martinez-Amat et al., 2013 [31]	No warming-up exercises were performed. Developing: Consisted of three stages (initial, intermediate and advanced) progressing to static and dynamic exercises. Six exercises of hip and knee on a firm surface, then on a soft surface (BOSU ball), were included (30 min). Cool-down: 10 min with slow walk, mobility and stretching exercises.
Resende Neto et al., 2017 [32]	Warm-up: joint mobility. Developing: intermittent activities organized in a circuit that required agility, coordination, velocity and muscle power of a set of complex motor systems. In addition, they included multi-articular exercises for lower and upper limbs with intense activation of stabilizing muscles of the spine, organized in a circuit and intermittent high-intensity activities (60 min). Cool-down was not reported.
Rezaeipour & Apanasenko, 2020 [33]	Warm-up: shuttle run, backward running, five stretching techniques for the trunk and lower extremities, two strengthening exercises, two impact-training techniques, such as box jumps, as well as doing two-leg versus one-leg jumping exercises. Agility: walk and run in different directions. (20 min). Developing: weight training, based on the recommendations of the American Heart Association and the American College of Sports Medicine (45 to min). Cool-down: activities not reported (10 min).
Stolzenberg et al., 2013 [34]	Warm-up: cycle ergometry (15 min). Developing: one set of training using standard gym equipment. Finally, neuromuscular training including Romberg, Tandem and Single Leg Stance were performed on surfaces of varying degrees of instability and with varying degrees of difficulty: firm mat, soft mat, wobble board, air-pillows, with and without shoes, with eyes open or with eyes closed. Softballs, tennis balls, staves and elastic bands were used for coordination training. Warm-up and cool-down were not reported.
Tsauo et al., 2008. [35]	Developing: sling suspension systems (TerapiMaster, Nordisk Terapi AS, Norway), where exercises in the supine, sitting and standing positions were performed sequentially. Warm-up and cool-down were not reported. * Time was not reported.

HRmax.: maximum heart rate.

## Data Availability

The data that support the findings of this study are available from the corresponding author upon reasonable request.

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
