# Peer review of "Effects of Neuromuscular Training on Physical Performance in Older People: A Systematic Review"

_life, 2023, doi:10.3390/life13040869_

Round 1

Reviewer 1 Report

The study is focused on the effects of neuromuscular training on physical performance in older people. This is a very interesting topic to search possibilities to find useful methods to keep patient in good health. Methodology is typical. Please add exclusion criteria. Paper is well written with the clear figures and tables. Text is clear and easy to read. Limitations included. Please add information about injury prevention because it is stated in the material you reviewed.

Author Response

Dear reviewer, thank you for your comments. Regarding the request for the inclusion of exclusion criteria, we inform you that these are found within the text. I detail them below:
On the other hand, excluded records were: (i) cross-sectional, retrospective, and prospective studies, or whose interventions were not centered on neuromuscular training; (ii) non-original articles (e.g., letters to the editor, translations, notes, book reviews); (iii) duplicate articles; (iv) review articles (e.g., meta-analyses, systematic reviews, narrative reviews); and (v) case studies.   In addition, the request on injury prevention in older people was incorporated.  

Reviewer 2 Report

This systematic review seems interesting and provides useful information about the role of Neuromuscular Training on Physical Performance in Older People. The manuscript is written well and acceptable for publication but need minor revision.

1)     Manuscript language can be improved.

2)     The manuscript needs minor correction, spelling, spacing etc.

3)     Endnote should be used for citation and reference.

Author Response

Dear reviewer, thank you for your comments. We reviewed spelling, spacing and language of the manuscript. Regarding the use of Endnote, we use another bibliographic manager (Zotero), which has the same function as the endnote program. In addition, we carry out a manual review of the bibliographic references according to the norms established by the journal.
